# The Applicability of Artificial Intelligence in Predicting the Depth of Myometrial Invasion on MRI Studies—A Systematic Review

**DOI:** 10.3390/diagnostics13152592

**Published:** 2023-08-03

**Authors:** Octavia Petrila, Anca-Elena Stefan, Dumitru Gafitanu, Viorel Scripcariu, Ionut Nistor

**Affiliations:** 1Faculty of Medicine, University of Medicine and Pharmacy “Grigore T. Popa”, 700115 Iasi, Romania; octavia.ciuhodaru@gmail.com (O.P.); dgafit@yahoo.com (D.G.); viorel.scripcariu@umfiasi.ro (V.S.); ionut.nistor@umfiasi.ro (I.N.); 2Department of Radiology, “Sfantul Spiridon” Hospital, 700111 Iasi, Romania; 3Department of Nephrology, “Dr. C.I. Parhon” Hospital, 700503 Iasi, Romania; 4Department of Obstetrics and Gynecology, “Elena Doamna” Hospital, 700398 Iasi, Romania; 5Regional Institute of Oncology, 700483 Iasi, Romania

**Keywords:** endometrial cancer artificial intelligence, endometrial cancer AI, endometrial cancer MRI artificial intelligence, endometrial cancer machine learning, endometrial cancer machine learning MRI

## Abstract

(1) Objective: Artificial intelligence (AI) has become an important tool in medicine in diagnosis, prognosis, and treatment evaluation, and its role will increase over time, along with the improvement and validation of AI models. We evaluated the applicability of AI in predicting the depth of myometrial invasion in MRI studies in women with endometrial cancer. (2) Methods: A systematic search was conducted in PubMed, SCOPUS, Embase, and clinicaltrials.gov databases for research papers from inception to May 2023. As keywords, we used: “endometrial cancer artificial intelligence”, “endometrial cancer AI”, “endometrial cancer MRI artificial intelligence”, “endometrial cancer machine learning”, and “endometrial cancer machine learning MRI”. We excluded studies that did not evaluate myometrial invasion. (3) Results: Of 1651 screened records, eight were eligible. The size of the dataset was between 50 and 530 participants among the studies. We evaluated the models by accuracy scores, area under the curve, and sensitivity/specificity. A quantitative analysis was not appropriate for this study due to the high heterogeneity among studies. (4) Conclusions: High accuracy, sensitivity, and specificity rates were obtained among studies using different AI systems. Overall, the existing studies suggest that they have the potential to improve the accuracy and efficiency of the myometrial invasion evaluation of MRI images in endometrial cancer patients.

## 1. Introduction

Endometrial cancer is the second most common gynecological cancer worldwide, with an increasing incidence in high-income countries. Imaging evaluation using magnetic resonance imaging (MRI) plays a crucial role in treatment planning, as it provides information on tumor staging (including the size and depth of invasion in the myometrium and cervical stroma but also in pelvic anatomical structures) and lymph node status. For endometrial cancer staging, the International Federation of Gynecology and Obstetrics (FIGO) uses a surgical staging system, but expert opinions and contemporary studies recommend MRI evaluation pre-treatment to choose the most appropriate therapy [1,2,3].

The most important morphological factor that affects the prognosis of these patients is the depth of myometrial invasion. Lymph node metastases are more frequent (46%) in cases of profound invasion than in cases of superficial invasion (3%). In addition, by combining the depth of myometrial invasion with tumor grading, histologic type, and tumor volume, we can clearly stratify the risk of recurrence and overall survival for these patients [4,5,6].

The ability to determine preoperative MRI stages based on personal expertise and experience can vary dramatically among individuals [7,8]. Moreover, staging can be affected by other pathological factors such as adenomyosis or leiomyomas. Taking all these factors into account, the differences between preoperative MRI staging and pathological diagnosis can differ. Introducing artificial intelligence (AI) assistance in endometrial cancer diagnosis can minimize these differences. Radiologists have started using artificial intelligence to read medical images of various diseases; however, the use of this method in endometrial cancer MRI images is rare. Few studies have addressed this issue, and the actual benefits remain questionable. In critical cases, the radiologist tends to be more cautious and may recommend other imaging investigations or interventions, but AI might be clearer and more precise [9,10,11].

Although the current AI technology may not be able to replace the expertise and experience of physicians, it can be used as an auxiliary resource. Having a “second opinion” can be helpful for radiologists, especially in critical cases. Accurate diagnosis followed by appropriate treatment in the early stages is key to a good prognosis. Currently, MRI is the primary tool used to assess the depth of myometrial invasion before surgery [10,11].

We conducted a systematic review of the impact of using AI systems to evaluate the depth of myometrial invasion on MRI images in endometrial cancer patients.

## 2. Materials and Methods

The updated Preferred Reporting Items for Systematic Reviews and Meta-Analyses (PRISMA) guidelines were applied to standardize data search, collection, synthesis, and reporting. For our systematic review, we used a protocol registered at https://osf.io (accessed on 14 June 2023) (DOI 10.17605/OSF.IO/XC6TR).

### 2.1. Data Sources and Search Strategy

We conducted a comprehensive search of several databases from inception to May 2023. The databases included PubMed, SCOPUS, Embase, and ClinicalTrials.gov. A hand-search of relevant radiology journals was also performed. We used different combinations of keywords and controlled vocabulary to create a comprehensive search strategy: “endometrial cancer artificial intelligence”, “endometrial cancer AI”, “endometrial cancer MRI artificial intelligence”, “endometrial cancer machine learning”, and “endometrial cancer machine learning MRI”. Language filters were not applied during the search process. The complete search strategy is described in Appendix A.

### 2.2. Study Selection

Two reviewers (OP and AS) independently screened the titles and abstracts; if considered eligible, full-text articles were assessed independently. The inclusion criteria were publications dated from inception to May 2023. Eligible studies included retrospective and prospective studies evaluating the success of AI systems in establishing the depth of myometrial invasion compared with radiologists and pathological results. Preclinical studies, duplicate data, study protocols, systematic or narrative reviews, meta-analyses, letters, commentary, editorials, surveys, guidelines, and recommendations were excluded. Furthermore, studies focusing on other types of gynecological cancers were excluded.

The qualitative synthesis of the results was provided using a narrative approach.

### 2.3. Data Extraction

Data obtained from studies were (1) author, (2) year of publication, (3) country, (4) type of study, (5) number of patients included, (6) clinical parameters, (7) endometrial cancer stage, (8) associated myometrial pathology, (9) input data, (10) radiomics, (11) referral region, (12) MRI machine, (13) output, (14) AI model, (15) segmentation, (16) number of patients used for training, (17) accuracy, (18) AUC, (19) sensitivity/specificity. Depending on the study design, the proper evaluation of the prediction performance or the robustness of the model is controversial. We extracted data regarding accuracy score, AUC, and sensitivity/specificity.

We assessed the risk of bias by applying the prediction model risk of the bias assessment tool (PROBAST). The risk of bias was classified as low, moderate, or high. The evaluation tool contained 20 signaling questions from four domains: participants, predictors, outcomes, and analyses.

## 3. Results

Our initial search identified 2665 articles: 1654 on Medline, 685 on Embase, and 326 on SCOPUS. After removing duplicates, 1651 titles and abstracts were examined, of which 1600 were excluded based on title or abstract. A total of 51 articles were retrieved and read in full. Articles that did not meet the inclusion criteria, specifically those that did not evaluate myometrial invasion, were excluded. Among the articles screened, some discussed the usefulness of AI in diagnosing endometrial cancer; however, there was no mention of myometrial invasion. Two of the retrieved articles were systematic reviews and meta-analyses and were excluded. Thus, eight articles were included in our study with a total number of 1543 participants. Figure 1 shows the PRISMA flowchart of this systematic review.

As shown in Table 1 (Characteristics of studies), the number of publications regarding the subject increased from 2020 onwards. All studies were retrospective. Input data were obtained from imaging studies using an MRI.

Only one of the included studies had associated myometrial pathology [9]. Regarding endometrial cancer stage, in four studies, participants had stage I endometrial cancer; stage I is the only one that is defined by the depth of myometrial invasion (Ia < 50% of the myometrial thickness affected, Ib > 50% of the myometrial thickness affected); in stages II, III, and IV, the tumor affects more than the uterine corpus, without taking into consideration the depth of myometrial invasion [19]. The follow-up period among studies varied from 3 years to 13 years. Three studies used 1.5T MRI machines, one study used 3T MRI machines, three studies used both 1.5T and 3T MRI machines, and one study did not report the type of MRI used. The preferred sequence used among studies was sagittal T2w, with or without fat suppression. Diffusion and post-contrast are the most frequently used sequences. In five of eight studies, image segmentation is manual based on the experience of the radiologist.

As seen in Table 2 (Applicability of different AI models), the depth of myometrial invasion was evaluated among all studies. In five of eight studies, myometrial lesion was the region of interest. Used artificial intelligence models were either machine learning or deep learning models, with five studies also adding radiomics. Accuracy, AUC, sensitivity, and specificity were calculated among the studies.

Chen et al. developed a deep learning model based on convolutional neural networks for the automatic identification of the endometrial lesion and appreciation of the depth of myometrial invasion. The model obtained an accuracy of 84.8%, a sensitivity of 66%, and a specificity of 87.5%. The radiologist obtained an accuracy score of 78.3%, a sensitivity score of 61.1%, and a specificity score of 80.8%. However, the best results were obtained when the radiologist collaborated with the software (accuracy of 86.2%, sensitivity of 77.8%, and specificity of 87.5%) [12].

Dong et al. developed a model that obtained a similar diagnostic accuracy to the radiologist (79.2% using T1 post-contrast sequence and 70.8% using T2w compared to 77.8% obtained by the radiologist). This study showed that the closer the invasion is to the threshold value of 50%, the more difficult to diagnose it becomes for both AI and the radiologist. Dong et al. is also the only study that evaluated the influence of leiomyomas when evaluating the depth of myometrial invasion. AI software seems influenced by the presence of leiomyomas when compared to the radiologist. [8] The one thing that was not taken into consideration in this study was the position of the leiomyomas. We know that intrauterine leiomyomas can develop in three main locations: intramural (most common as it develops within the myometrium), submucosal (the least common, projecting into the uterine cavity), and sub-serosal (the kind that projects outside the uterus). Only the first two types of leiomyoma can affect the evaluation of the depth of myometrial invasion, as they are the only two that modify either the uterine cavity and the endometrial line or the myometrial thickness [20].

Mao et al. is the only study whose main purpose was the early staging of endometrial cancer in stage Ia or Ib, meaning superficial or profound myometrial invasion. This study did not use radiomic or textural criteria. It used volumetric criteria, and the best results were obtained in T2 sagittal (diagnostic accuracy of 0.914, sensitivity of 0.923, and specificity of 0.909) [13].

Otani et al. used the radiomic analysis of the images not only to determine myometrial invasion but also to determine histologic grading, lympho-vascular invasion, and metastasis in pelvic and paraaortic nodules. Regarding myometrial invasion, the results were similar to the radiologists that analyzed the images. Moreover, when radiologists received “help” from the AI software, no improvement in appreciating myometrial invasion was observed [14].

Using GLCM (gray level co-occurrence matrix), Qin et al. identified radiomic elements that differentiated stage Ia from Ib. More models were used then (RFC, SVM, XGBoost, ANN, and DT) in order to appreciate myometrial invasion, obtaining an AUC between 0.765 and 0.8777 in the training set and between 0.716 and 0.862 in the test set [15].

Rodriguez et al. combined textural analysis on T2 sequence with ADC map and semi-quantitative maps derived from post-contrast sequences in order to determine the depth of myometrial invasion. Results showed a sensitivity of 80.95% and specificity of 93.33%. When taken individually, the same parameters showed lower sensitivity and specificity rates [16].

The depth of myometrial invasion is difficult to appreciate when there are voluminous tumors that distend the uterine cavity and thin the uterine wall. Stanzione et al. wanted to limit these shortcomings and developed a model that obtained an accuracy score of 0.92 in the training set and 0.94 in the test set [17].

Zhu et al. is the only study that proposed a computer-aided diagnostic (CAD) variant in order to appreciate the depth of myometrial invasion using only the uterine segmented region. Moreover, a geometric feature (LS) was proposed to determine the irregularity of the tissue structure. After the textural analysis and integration of the selected parameters, a model called EPSVM was developed. It obtained an accuracy of 93.7%, a sensitivity of 94.7%, and a specificity of 93.3%. Using only the geometric feature, accuracy was 89.9%, sensitivity was 89.5%, and specificity was 90% [18].

We summarized the risk of bias for each model (Table 3) using PROBAST. In the domains of participants, predictors, and outcomes, most studies were classified as low risk. However, in the domain of analyses, most studies were classified as high risk owing to the number of participants included in the analysis. Moreover, the number of participants and lack of external validation were limitations of several studies.

## 4. Discussion

This review evaluated the applicability of AI systems in predicting the depth of myometrial invasion on MRI studies in women with endometrial cancer. It included eight eligible articles, and the dataset among studies was between 50 and 530. A different AI system was used in each study included.

The use of AI systems to evaluate the depth of myometrial invasion in MRI images in endometrial cancer patients can have a significant impact on clinical practice. Myometrial invasion is a crucial factor in determining the stage and treatment plan for endometrial cancer patients, and accurate assessment is essential for optimal patient care.

AI systems can provide a more objective and standardized approach to evaluating myometrial invasion in MRI images. This can lead to improved accuracy and consistency in diagnosis, staging, and treatment planning, which can ultimately result in better patient outcomes. Furthermore, AI systems can analyze large amounts of data quickly, allowing for more efficient and timely evaluations of myometrial invasion. This can lead to the earlier diagnosis and treatment of endometrial cancer, which is critical for improving survival rates and reducing morbidity.

While the automatic diagnosis of any neoplasia is far away, real results that we want in an AI system are the characterization and recognition of some lesions using quantitative methods. Up until now, there have been considerable variations between radiologists and even between a radiologist from one evaluation to the other. The current systematic review showed good results of different AI models in appreciating the depth of myometrial invasion.

One limitation of our study is that the AI systems used throughout the included studies are not standardized. Five of the studies used machine learning, and three studies used convolutional neural networks. Another limitation is that although all models obtained satisfactory results, in order to verify the efficacy of an AI system, a larger number of participants and images are needed. Three of the studies had more than 200 participants. Of the participants included, more than half were used for the training set.

When talking about reference images used for training and testing the software, these were processed in two studies. All studies used “ideal” images, with optimal contrast and without artefacts that make them perfect for analysis and interpretation. This is hard to obtain in the current practice when the quality of imaging is not always “ideal”.

One study (Dong et al. [9]) analyzed the influence of leiomyomas when appreciating the depth of myometrial invasion. The study showed that the model they used was not as accurate when leiomyomas were involved, while the radiologists’ diagnosis was not as influenced by the presence of leiomyomas. We consider this observation vital, given the fact that, in day to day practice, we encounter females with associated uterine pathology (adenomyosis, polyposis, etc.).

In four studies, imaging segmentation was performed manually, which means that the drawn contours of the uterus and endometrium can vary depending on the radiologist’s experience. Then, the AI system extracts data based on these markings. The manual segmentation of imaging can predispose variations in the diagnostic accuracy of the software. The automatic segmentation of the uterus is easier to obtain, given the fact that its shape is geometric and stable, but problems can appear in the tumoral automatic segmentation due to variable shapes and different shades of gray included in the same lesion. However, there are factors that can influence the automatic segmentation of the entire image, such as the variable form of the tumors, intraperitoneal fluid, distension of the uterine cavity, or uterine leiomyomas.

Classical imaging diagnosis is based on the different shades of gray included in the same image, which makes it subjective. This is why radiomics was developed. An AI software operates with vectors, the shorter the vector the more effectiveness is gained; radiomics is important in order to create these vectors with which AI software can operate. One disadvantage is that the elements extracted using radiomics substitute the baseline image, excluding essential elements that can be important for radiological analysis. Five studies used radiomics for the appreciation of myometrial invasion, but each study used different radiomics criteria.

## 5. Conclusions

Artificial intelligence can be a good help in appreciating the depth of myometrial invasion, but research in this area is in development. Overall, while there is still a need for further research to validate the use of AI systems in clinical practice, the existing studies suggest that they have the potential to improve the accuracy and efficiency of myometrial invasion evaluation in MRI images in endometrial cancer patients.

## Figures and Tables

**Figure 1 diagnostics-13-02592-f001:**
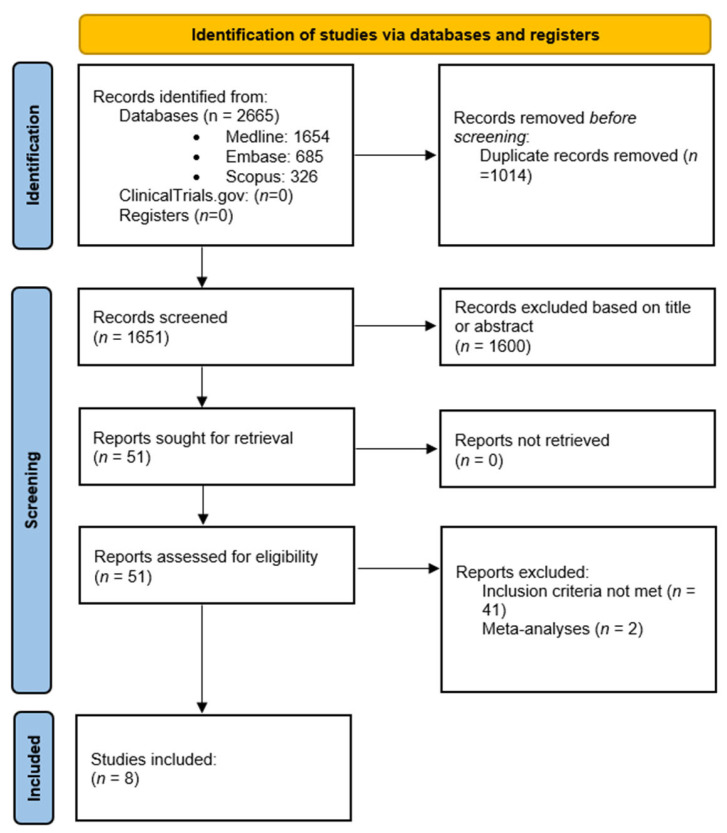
PRISMA flowchart of this systematic review.

**Table 1 diagnostics-13-02592-t001:** The characteristics of all included studies.

Author/Year	Country	Type of Study	No. Patients	Clinical Parameters	Endometrial Cancer Stage	Myometrial Pathology	Follow-Up
Chen et al., 2020 [12]	China	Retrospective	530	Age, menopausal status, BMI, stage, grade	I, II, III, IV	No	5 years
Dong et al., 2020 [9]	Taiwan	Retrospective	72	Age, menopausal status, ECOG performance	I	Yes	4 years
Mao et al., 2022 [13]	China	Retrospective	117	Age	I	No	3 years
Otani et al., 2022 [14]	Japan	Retrospective	200	n/a	I, II, III, IV	No	13 years
Qin et al., 2022 [15]	China	Retrospective	348	Age, menopausal status, drinking, smoking, HTN, BMI, diabetes, CA19.9, CA125	I	No	10 years
Rodriguez et al., 2021 [16]	Spain	Retrospective	143	Age, FIGO, depth, lympho-vascular space invasion	I, II, III, IV	No	6 years
Stanzione et al., 2020 [17]	Italy	Retrospective	54	Age	IA, Ib, II, IIIa	No	3 years
Zhu et al., 2021 [18]	China	Retrospective	79	Age	I	No	3 years

**Table 2 diagnostics-13-02592-t002:** Applicability of different AI models.

Author/Year	Input Data	Radiomics	Referral Region	MRI Machine	Output	AI Model	Segmentation	No. Patients Used for Training	Accuracy	AUC			Sensitivity/Specificity				
Chen et al., 2020 [12]	Imaging data (MRI)	No	Lesion	1.5T	DMI	DL (CNN)	Automated	313	Radiologist: 78.3%	Train: 0.85			ML: SENS 66.7%, SPEC: 87.5%				
									ML: 84.8%	Validation: 0.81			Radiologist + ML: SENS: 77.8%, SPEC: 87.5%				
									Radiologist + ML: 86.2%	Test: 0.78							
Dong et al., 2020 [9]	Imaging data (MRI)	No	Uterus, endometrium, lesion	1.5T and 3T	DMI	CNN	Automated	24	Radiologist: 77.8%	n/a			T1W (ML)		SPEC:85.9%	SPEC 73.1%	SPEC: 84.3%
									AI: T1w: 79.2%				T2W (ML)		SPEC:83.6%	SPEC:82.8%	SPEC: 87.3%
Mao et al., 2022 [13]	Imaging data (MRI)	No	Uterus, lesion	1.5T	Staging Ia/Ib (TUR)	CNN	Manual, then automated	70	Axial T2Wi: 0.857	Axial T2wi: 0.86			Axial T2WI: SENS:0.846, SPEC: 0.864				
									Axial DIWI: 0.857	Axial DWI: 0.85			Axial DIWI: SENS: 0.692, SPEC: 0.955				
									Sagittal T2Wi: 0.914	Sagittal T2WI: 0.94			Sagittal T2Wi: SENS 0.923, SPEC: 0.909				
Otani et al., 2022 [14]	Imaging data (MRI), age, CA125, CA19-9	Yes	Lesion	1.5T and 3T	DMI, histological grade, lympho-vascular invasion, pelvic/paraaortic lymph node metastasis	ML	Manual	150	Discovery dataset: 0.65	Discovery dataset: 0.76				Reader 1	Reader 2	Reader 3	Reader 4
									Test dataset: 0.76	Test dataset: 0.83				Before/After ML			
													SENS	0.61/0.610	0.77/0.72	0.72/0.72	0.66/0.66
													SPEC	0.96/0.96	0.78/0.87	0.87/0.87	0.9/0.87
Qin et al., 2022 [15]	Imaging data (MRI)	Yes, GLCM	Lesion	n/a	DMI	ML (RFC, SVM, ANN, DT, XGBoost)	Manual	70%	N/A		Training	Test	N/A				
										RFC	0.877	0.862					
										SVM	0.765	0.716					
										DT	0.787	0.739					
										ANN	0.842	0.804					
										XGBoost	0.768	0.715					
										Radiologist	0.835	0.816					
Rodriguez et al., 2021 [16]	Imaging data (MRI)	Yes	Lesion	1.5T and 3T	DMI	ML	Manual	107	T2W Texture: 58.33	T2W Texture: 59.38			T2W Texture: SENS: 50, SPEC: 68.75				
									ADC Texture: 63.89	ADC Texture: 63.13			ADC Texture: SENS: 70, SPEC: 56.25				
									DCE description: 42.86	DCE description: 41.43			DCE description: SENS: 42.86, SPEC: 40				
									T2W texture: ADC+ DCE:ADC descriptors: 86.11	T2W texture: ADC+ DCE:ADC descriptors: 87.14			T2W texture: ADC+ DCE:ADC descriptors: SENS: 80.95, SPEC: 93.33				
Stanzione et al., 2020 [17]	Imaging data (MRI)	Yes	Lesion	3T	DMI	ML	Manual	80%	Training: 86%	Training: 0.92			Training: SENS 0.71, SPEC: 0.93				
									Test: 91%	Test: 0.94			Test: SENS: 0.67, SPEC: 1.00				
Zhu et al., 2021 [18]	Imaging data (MRI)	Yes	Uterus	1.5T	DMI	EPSVM	Automated	All	EPSVM 93.7%	EPSVM: 0.922			EPSVM: SENS: 94.7%, SPEC: 93.3%				

DMI: deep myometrial invasion, AUC: area under the curve; SENS: sensitivity, SPEC: specificity; ML: machine learning; RFC: random forest classifier, SVM: support vector machine, DT: decision tree, ANN: artificial neural network, XGBoost: extreme gradient boosting, ADC: apparent diffusion coefficient, DCE: dynamic contrast-enhanced, EPSVM: support vector machine.

**Table 3 diagnostics-13-02592-t003:** Risk of bias using PROBAST method.

Study	ROB				Applicability			Overall	
	Participants	Predictors	Outcome	Analysis	Participants	Predictors	Outcome	ROB	Applicability
Chen et al., 2020 [12]	+	−	+	−	+	+	+	+	+
Dong et al., 2020 [9]	+	−	−	−	+	+	-	+	−
Mao et al., 2022 [13]	+	+	+	−	+	+	+	+	+
Otani et al., 2022 [14]	+	−	+	+	+	Unclear	+	+	+
Qin et al., 2022 [15]	+	+	+	−	+	−	−	+	+
Rodriguez et al., 2021 [16]	+	−	+	+	+	−	−	+	+
Stanzione et al., 2020 [17]	+	−	+	−	+	Unclear	−	+	−
Zhu et al., 2021 [18]	+	+	+	−	+	Unclear	+	+	−

## Data Availability

Not applicable.

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
