# Peer review of "The Applicability of Artificial Intelligence in Predicting the Depth of Myometrial Invasion on MRI Studies—A Systematic Review"

_diagnostics, 2023, doi:10.3390/diagnostics13152592_

Round 1

Reviewer 1 Report

The manuscript entitled "The applicability of artificial intelligence in predicting the depth of myometrial invasion on MRI studies –a systematic view" is study the the AI have potential to improve the accuracy and efficiency of myometrial invasion evaluation on MRI images in endometrial cancer patients. Abstract is covered main contents. Background is good arranged with sufficient literature review. Methods are routine and logical. Statistical tests are correct. Results show findings clearly. Conclusion is appropriate. References are related to the issue and up-to-date. I think the figures and tables are important and help to better understanding of the subject.

I think this manuscript was well organized and it could be accepted.

The manuscript entitled "The applicability of artificial intelligence in predicting the depth of myometrial invasion on MRI studies –a systematic view" is study the the AI have potential to improve the accuracy and efficiency of myometrial invasion evaluation on MRI images in endometrial cancer patients. Abstract is covered main contents. Background is good arranged with sufficient literature review. Methods are routine and logical. Statistical tests are correct. Results show findings clearly. Conclusion is appropriate. References are related to the issue and up-to-date. I think the figures and tables are important and help to better understanding of the subject.

I think this manuscript was well organized and it could be accepted.

Author Response

Thank you for your review. We appreciate your support for our manuscript and attached the response to the comments.

Reviewer 2 Report

''The applicability of artificial intelligence in predicting the depth of myometrial invasion on MRI studies - a systematic review''

The manuscript (systematic review) consists of 12 pages with 20 references. The Authors focused on the issue concerning the use of artificial intelligence to assess the depth of myometrial invasion. The study is divided into sections: Introduction, Materials and Methods, Results, Discussion, Conclusions. (Section 6 ''Patents'' - mistakenly inserted?) 

The study fits the Journal scope - an artificial intelligence as an additional tool in diagnosis of endometrial cancer. The subject of the study is actual and relevant. 

The aim of the study is defined - in line 62-63.

Materials and Methods:

-The study is well-structured. PRISMA guidelines were described in detail. The selection of the articles was clearly presented. 

Results:

-In Abstract in results it is written that''1896'' articles were selected. In section Results it is mentioned that ''Our initial search identified 2665 articles'' - Could you clarify this information? 

Discussion:

- ''the dataset was between 50 and 500'' - line 193-194. In abstract in line 22 ''between 50 and 550'' - Could you clarify this information?

- Conclusions in line 201 and 207  seem to be the same (repetition).

References:

-Could you write literature according to guidelines?

1. Author 1, A.B.; Author 2, C.D. Title of the article. Abbreviated Journal Name YearVolume, page range.

- 3,7,8,9,11,12,13,14,15,16,17,20 - title, year (in reverse order)

-repetition of words (for example line 219 ''included'')

Author Response

We thank you for your review. We have attached the point-by-point responses to the comments.
